# S-Former: Structural Anchoring for Stable Long-Context Modeling

## Abstract

Transformers form the backbone of modern large language models, but their long-context performance is limited by the *dilution effect*: attention mass spreads uniformly across distant positions, failing to maintain structural dependencies. Existing solutions, such as sparse or efficient attention patterns, improve efficiency but do not address the lack of *structural anchoring*. We introduce the **Structural-Former (S-Former)**, which maintains a parallel structural stream that evolves recurrently to track sequential patterns independently of token content and provides structural-like anchors for attention. Unlike compressed state-space models, our approach maintains explicit structural representations that remain orthogonal to semantic content. We study two integration mechanisms: (i) **attention fusion**, which validates the decoupling principle by showing that the structural gate $\alpha_t$ tracks bracket depth in Dyck languages; and (ii) **bias injection**, a minimal and stable design that adds the structural signal into hidden activations. Synthetic probes (Markov, Dyck and JSON) demonstrate that the structural stream learns hierarchical and sequential rules beyond surface statistics. On WikiText-103, S-Former extrapolates stably to long contexts, **reducing perplexity degradation by 76% when extrapolating to 40k tokens**. These findings suggest that introducing a recurrent structural stream provides a lightweight and scalable *inductive bias* that substantially improves long-context extrapolation, offering a complementary direction to sparse attention or memory-based methods.

## 1 Introduction

Transformers have become the backbone of large language models, powering advances in language understanding, and reasoning. Their success stems from self-attention, which models token interactions flexibly. However, attention encodes order only in a *metric-based* fashion—via positional indices or rotary embeddings—without explicitly modeling the **underlying structure** of sequences. Natural language, code, and music are not linear token chains but follow recursive and hierarchical patterns. These signals are inherently twofold:

- **Content**, carrying semantic meaning;
- **Structure**, defining how elements are arranged.

Standard Transformers conflate content and structure, forcing the latter to be inferred indirectly from token distances. As sequence length grows, attention mass becomes diffuse, producing the *dilution effect* and weakening long-range dependencies.

We address this limitation by introducing a **structural stream**: a recurrent pathway that tracks sequential dynamics and provides **structural anchors** for attention. Unlike memory caches, it evolves smoothly with the sequence, offering **stability** without explicit storage. To integrate this stream, we propose a lightweight **bias-injection mechanism**, which adds structural signals into hidden activations. This design minimally interferes with content while substantially improving long-context extrapolation—remaining stable up to 40k tokens with negligible overhead.

Our approach highlights a broader insight: **long-context stability does not require heavy memory or architectural overhauls**. Even a lightweight structural stream is sufficient to anchor continuity across tens of thousands of tokens. Crucially, **our design is not a memory cache.** Unlike episodic or compressed memory systems, the structural stream **evolves continuously with the sequence**,

providing stable anchors without explicit storage. Empirically, it shows **high-rank utilization, orthogonality to content, and smooth temporal dynamics**, supporting its role as a **lightweight inductive bias for long-range stability** rather than external memory.

Our validation proceeds in three stages. (1) On **synthetic tasks**, we show that the stream captures hierarchical patterns standard Transformers miss. (2) Using **attention fusion** as a probe, we obtain interpretable $\alpha$–structure correlations, though with stability limitations. (3) We then introduce **bias injection** as a practical design, which trades some interpretability for stable, scalable long-context modeling up to 40k tokens.

**Contributions.** (1) We propose a **structural anchoring framework** where a recurrent stream stabilizes long-context modeling. (2) Through **synthetic probes**, **attention fusion**, and a scalable **bias-injection design**, we show that the stream learns hierarchical patterns. (3) We provide **theoretical and empirical evidence** that it induces structural representations rather than a memory cache, offering a lightweight inductive bias for extrapolation.

## 2 RELATED WORK

**Transformers, Positional Encoding, and Instability.** The Transformer architecture (Vaswani et al., 2017) underpins modern large language models, with positional encodings (absolute, relative, RoPE (Su et al., 2021), and ALiBi (Press et al., 2022)) providing distance-aware signals for otherwise permutation-invariant attention. These methods remain fundamentally *metric-based*: they encode relative offsets but not higher-order sequential regularities. Analyses further show that purely metric encodings lead to *dilution* and instability in long contexts, where attention mass spreads uniformly and weakens continuity (Liu et al., 2020; Xiong et al., 2020; Elhage et al., 2021).

**Memory-Augmented and Recurrent Models.** Another line of work extends context through recurrent or state-space dynamics, such as RWKV (Peng et al., 2023), RetNet (Sun et al., 2023), and Mamba (Gu & Dao, 2024). These architectures focus on efficiency and horizon length, but their states are compressive summaries evolved in a generic manner. In contrast, our recurrent stream evolves independently of token content, maintaining disentangled structural dynamics rather than compressed memory.

**Structural Bias in Language.** Inductive biases have been explored through linguistic priors, such as tree-based attention (Shiv & Quirk, 2019), syntactic pre-training (Li et al., 2020), and Struct-Former (Shen et al., 2021). Recurrent models are also known to emulate formal devices such as finite-state machines and counters (Weiss et al., 2018; Merrill, 2019; Chiang & Siegelmann, 2020). Our approach differs in focusing on lightweight structural anchoring that operates continuously during training, without requiring explicit syntactic trees or symbolic supervision.

**Sparse and Efficient Attention.** Sparse and efficient attention mechanisms such as Longformer (Beltagy et al., 2020), BigBird (Zaheer et al., 2020), Performer (Choromanski et al., 2021), Linformer (Wang et al., 2020), and ETC (Ainslie et al., 2020) reduce quadratic cost via sparsity, low-rank approximation, or global tokens. Our approach is complementary: we preserve dense attention for accuracy, while targeting stability in extrapolation.

## 3 METHODS

### 3.1 PRELIMINARIES: METRIC-ONLY ATTENTION

We take as baseline a vanilla Transformer with rotary position embeddings (RoPE), which represents long-context modeling purely through metric distance. A standard Transformer layer computes self-attention as

$$A(Q, K) = \text{softmax}\left(\frac{QK^\top}{\sqrt{d}}\right), \quad O(Q, K, V) = A(Q, K)V, \tag{1}$$

where queries $Q$, keys $K$, and values $V$ are linear projections of input embeddings. This formulation encodes sequence order only via **positional embeddings** (absolute, relative, or rotary). As a result,

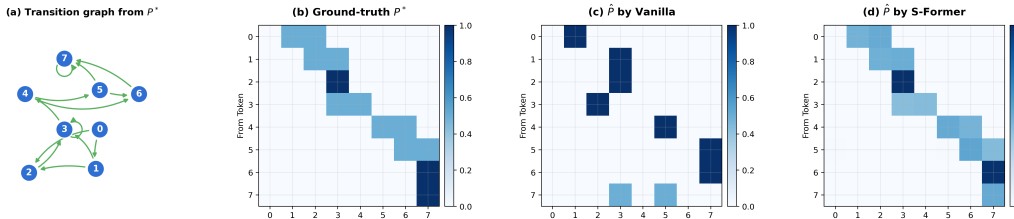

Figure 1: **Markov transition probe.** (a) Example state transition diagram. (b) Ground-truth transition operator. (c) Standard Transformer fails to recover the operator: attention weights become diffuse due to metric-only encoding. (d) **S-Former reconstructs the operator with high fidelity**, showing that the structural stream preserves sequential continuity and aligns closely with the true transition dynamics. This demonstrates that the structural stream acts as a **structural anchor** rather than a memory cache, enabling explicit modeling of sequential transitions.

order is represented as a *metric distance*, not as underlying **structural relations**. With increasing sequence length, attention scores tend to flatten, a phenomenon we call the **dilution effect**: probability mass spreads more uniformly across positions, weakening the model's ability to consistently retrieve structurally relevant anchors.

## 3.2 DECOUPLING CONTENT AND STRUCTURE VIA STRUCTURAL STREAM

We hypothesize that sequential data contains two distinct signals: **content**, which carries semantic meaning, and **structure**, which defines how elements are organized.

In practice, we realize structure through a recurrent stream that captures smooth temporal patterns and long-range dependencies, while maintaining near-orthogonality to content representations.

To capture this explicitly, we introduce a **structural stream**: a recurrent pathway running in parallel with attention. Concretely, given token embeddings $x_t$, we compute

$$g_t = \text{GRU}(g_{t-1}, \text{LN}(x_t)), \tag{2}$$

where $g_t$ is the **structural state**. We instantiate the structural stream with a GRU as a representative gated recurrent mechanism. Gated recurrence provides stable updates with data-dependent gating and integrates smoothly with Pre-LN Transformers.

Unlike attention, which re-computes relations at every step, the structural stream evolves persistently over time, acting as a **structural anchor** that guides the model toward long-range consistency while remaining decoupled from semantic content.

### 3.2.1 ATTENTION FUSION: DEMONSTRATING STRUCTURAL DECOUPLING

We introduce attention fusion as a probe to validate structure–content decoupling, rather than a final implementation. The structural state $g_t$ directly modulates the Query and Key projections, providing a dynamic balance between structural and content signals. Given token embeddings $x_t$ and the structural state $g_t$ (Section 3.2), we define:

$$\alpha_t^{fusion} = \sigma(W[g_t, \text{LN}(x_t)]) \tag{3}$$

$$Q_t = (\alpha_t g_t + (1 - \alpha_t)\text{LN}(x_t))W_Q, \ K_t = (\alpha_t g_t + (1 - \alpha_t)\text{LN}(x_t))W_K, \ V_t = \text{LN}(x_t)W_V \tag{4}$$

$$A'(Q, K) = \text{softmax}\left(\frac{QK^\top}{\sqrt{d}}\right) \tag{5}$$

Here $\alpha_t^{fusion}$ denotes a structure–content tradeoff gate: it controls the relative weight of structural state $g_t$ versus content embedding $\text{LN}(x_t)$ in forming $Q_t, K_t$.

On synthetic tasks such as Dyck languages, this design significantly improves structural tracking. A key observation is that the learned gating weights $\alpha_t^{fusion}$ strongly correlate with structural com-

plexity (e.g., bracket depth), as illustrated in Figure 2. This provides direct interpretability: the model explicitly learns to balance structure and content.

However, in natural text, we observed a content–structure trade-off. Multiplicative fusion can interfere with content representation, while also suffering from gradient instability when $\alpha_t^{fusion}$ approaches boundary values. Thus, while attention fusion serves as a strong conceptual proof of structural decoupling, it is not optimal for large-scale training.

### 3.2.2 BIAS INJECTION: OPTIMIZING FOR STABILITY

To overcome these limitations, we propose bias injection, where the structural stream is added to hidden activations instead of directly fused into attention logits. For a hidden state $h$ (either input embedding $x_t$ or residual activation $h^{\text{resid}}$), we compute:

$$h^{\text{biased}} = \text{LN}(h + \alpha_t g_t), \quad \text{where } \alpha_t^{bias} = \sigma(W[g_t, \text{LN}(h)]). \tag{6}$$

The attention projections then follow:

$$Q_t = h^{\text{biased}} W_Q, \quad K_t = h^{\text{biased}} W_K, \quad V_t = h^{\text{biased}} W_V. \tag{7}$$

Here $\alpha_t$ serves as an injection strength, modulating how much of the structural state $g_t$ is added to the hidden activation $h$. Although both are denoted $\alpha_t$, the two gating roles are conceptually distinct: fusion uses $\alpha_t^{fusion}$ for content–structure balancing, while injection uses $\alpha_t^{bias}$ for bias scaling.

This bias injection avoids the pitfalls of multiplicative fusion: it ensures **numerical stability** (no gradient blow-up when $\alpha_t \to 0$ or 1); **reduced parameter coupling** ($W_Q, W_K, W_V$ adapt only to a stable, content-augmented signal); and **efficiency** (minimal overhead while preserving long-range guidance). Empirically, this design proves both simple and effective: bias-injected Transformers extrapolate stably to 40k tokens on WikiText-103 with negligible perplexity increase.

### 3.3 TRAINING AND REGULARIZATION

The model is trained with standard language modeling loss. To prevent $\alpha_t$ from saturating at 0 or 1, we add a confidence penalty:

$$\mathcal{L} = \mathcal{L}_{\text{LM}} - \lambda \, \mathbb{E}_t[\alpha_t(1 - \alpha_t)]. \tag{8}$$

Our empirical strategy proceeds in three stages: (1) Synthetic experiments (Section 4) demonstrate that the structural stream can internalize symbolic and hierarchical patterns. (2) Attention fusion (Section 4.2.1) provides interpretable validation of content–structure decoupling, though it suffers from instability. (3) Bias injection (Section 4.2.2) introduces a more stable and scalable implementation, preserving structural anchoring across long contexts.

## 4 SYNTHETIC EXPERIMENTS

We first validate whether the structural stream has indeed learned hierarchical patterns that standard Transformers fail to capture. Synthetic tasks allow us to isolate structural signals explicitly.

We begin with controlled synthetic tasks to probe whether the structural stream indeed learns sequential and hierarchical structure beyond metric distance. Synthetic probes have long been used to test compositional generalization and formal language learning in neural models (Suzgun et al., 2019; Hahn, 2020; Lake & Baroni, 2018; Keysers et al., 2020).

### 4.1 SYNTHETIC PROBES: DOES THE STRUCTURAL STREAM ACTUALLY LEARN STRUCTURE?

We adopt two representative families: (1) **Markovian state-tracking tasks**, which test whether the model can maintain consistent latent dynamics under stochastic transitions; and (2) **Dyck languages and JSON serialization**, which together evaluate hierarchical generalization. Dyck languages require tracking nested brackets and serve as a canonical test of structural recursion. Building on this, JSON serialization adds semantic constraints through key–value consistency, extending the Dyck setting from purely syntactic depth-tracking to joint syntax–semantics consistency. Together, these

tasks **isolate structural reasoning requirements from semantic content**, allowing us to directly evaluate the structural stream's role as a coordinate anchor.

#### 4.1.1 MARKOV TRANSITION PROBE

Sequences are generated from a Markov chain (Graves et al., 2016; Bai et al., 2018; Bengio et al., 1994; Mikolov et al., 2012) (Fig. 1). The task is to recover the transition operator: given a state, predict its $k$-step distribution. Vanilla Transformers fail because attention is purely metric-based, causing transition information to diffuse into scattered, inconsistent weights (Fig. 1c). In contrast, S-Former's recurrent stream preserves sequential continuity, enabling it to **closely approximates** the operator (Fig. 1d), closely matching ground-truth (Fig. 1b).

#### 4.1.2 DYCK AND JSON PROBES

We first evaluate on the Dyck language benchmark (Suzgun et al., 2019; Hahn, 2020), a canonical test of hierarchical generalization. Unlike surface-level metrics, Dyck tasks require models to track recursive dependencies, making them a strong diagnostic for structural learning. Standard Transformers can memorize shallow patterns but collapse under deeper nesting, with structural accuracy dropping below 0.9. In contrast, S-Former leverages the structural stream as an implicit counter, sustaining up to 0.93 accuracy even under length and depth extrapolation. To interpret this behavior, we use *attention fusion* as a probe. The learned gating values ($\alpha$) show strong correlation with bracket depth across tokens (e.g., $r = 0.866$, $p < 10^{-7}$ on the illustrated sequence; see Fig. 2), and we find similarly high correlations across other Dyck sequences, indicating that this phenomenon is robust.This $\alpha$–depth correlation provides direct evidence of structure–content decoupling.

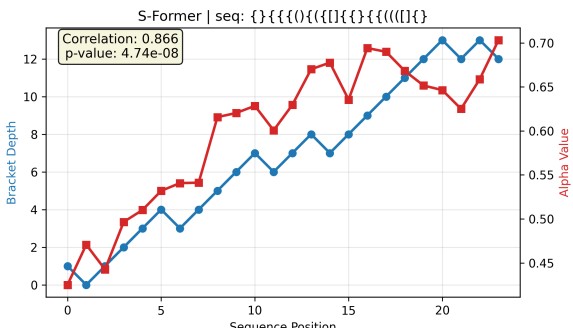

Figure 2: **Dyck language interpretability probe.** We use attention fusion as a conceptual probe to test whether the structural stream tracks hierarchical depth. The case study illustrates the correlation between the structural gating value $\alpha$ (red) and bracket depth (blue) within a single Dyck sequence. **S-Former shows a strong $\alpha$–depth correlation** (e.g., $r = 0.866$, $p < 10^{-7}$), providing direct evidence that the structural stream captures hierarchical structure.

Building on purely syntactic recursion, we further consider JSON serialization (Lake & Baroni, 2018; Keysers et al., 2020; Furrer et al., 2020; Herzig et al., 2021), which adds semantic constraints via key–value consistency. This extends the Dyck setting from syntax-only to joint syntax–semantics generalization. S-Former variants achieve up to 0.95 validity on held-out depths, representing the strongest result among our synthetic probes. For brevity, we report Dyck results in the main text and defer full JSON results to Appendix A.3.

Table 1: Structural accuracy on Dyck benchmark.

| Model | Struct acc |
|---|---|
| Standard Trans. (With PE) | 0.5620 |
| Standard Trans. (NoPE) | 0.7760 |
| RoPE Transformer | 0.5420 |
| **S-Former** | **0.9260** |

**Summary.** Across all probes, S-Former demonstrates that a structural stream provides the missing inductive bias: it tracks transitions, counts brackets, and maintains structural anchors that standard Transformers fail to represent. **This shows that long-context stability arises not from larger attention windows, but from structural anchoring that preserves continuity across tokens.**

### 4.2 NATURAL LANGUAGE MODELING: WIKITEXT-103

Having established the structural capacity of S-Former on synthetic probes, we now turn to natural corpora where the underlying structures are implicit. We evaluate on **WikiText-103**(Merity et al., 2016), a benchmark for long-form language modeling.

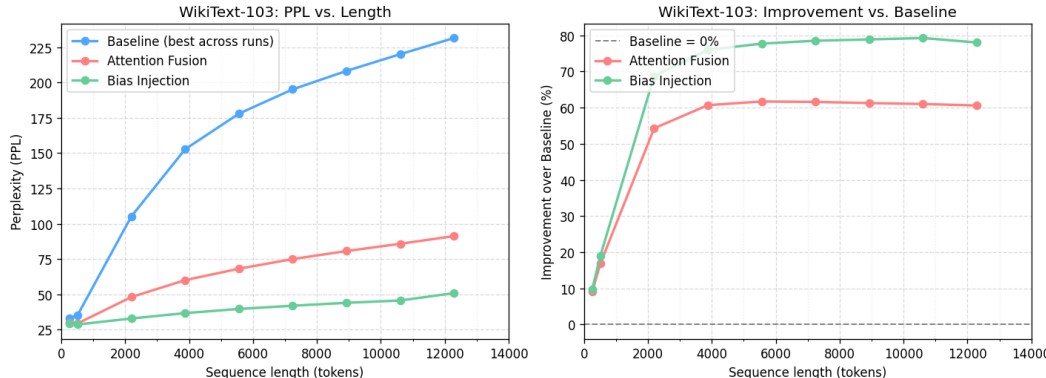

Figure 3: **Long-context extrapolation on WikiText-103.** We compare a vanilla Transformer baseline (RoPE) with two S-Former variants (attention fusion and bias injection). The plots illustrate how structural anchoring improves stability at longer contexts.

#### 4.2.1 ATTENTION FUSION: CAN WE PROVE STRUCTURE–CONTENT DECOUPLING?

After validating on synthetic probes, we next examine **attention fusion** as an interpretable probe. This mechanism reveals how $\alpha_t$ correlates with structural complexity, providing direct visualization of structure–content decoupling. However, it also exposes stability limitations when applied to natural text.

It is important to note that both baseline and S-Former were trained exclusively on 256-token windows. Nevertheless, when evaluated on longer sequences, S-Former yields significantly longer attention spans (Figure 4), whereas the baseline collapses to short-range focus. This indicates that the structural stream provides not just long-range memory, but a **structural inductive bias** that extrapolates beyond the training horizon.

When sequence length increases, a standard Transformer's perplexity degrades rapidly due to the *dilution effect*: attention mass becomes diffuse and fails to preserve continuity over long spans. For example, perplexity rises sharply from **32.8 → 231.3** as length extends from **256 → 12k tokens**. By contrast, S-Former with attention-fused structural streams remains significantly more stable: perplexity increases only from **30.2 → 91.22** over the same range. This yields a ~**46–58%** **relative improvement** across long contexts, with particularly pronounced gains beyond 2k tokens (e.g., **105.2 → 48.11 at length 2194; 220.0 → 85.8 at length 10k**).

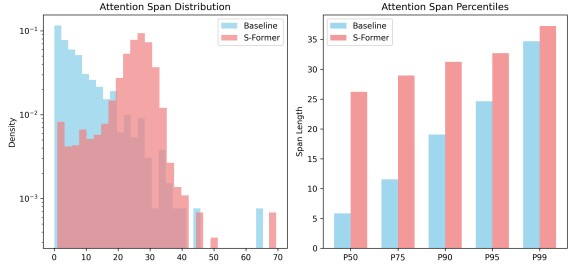

Figure 4: **Attention span analysis on WikiText-103 (attention fusion).** Both baseline and S-Former were trained on 256-token windows and evaluated on longer sequences. The figure illustrates how structural anchoring shifts attention toward longer spans, mitigating the dilution effect.

These results in Table 2 demonstrate that the structural stream consistently anchors long-range dependencies, even as vanilla attention collapses under extrapolation.

#### 4.2.2 BIAS INJECTION: HOW DO WE MAKE THIS PRACTICAL?

To overcome the instability of attention fusion while retaining structural guidance, we introduce **bias injection**. This additive mechanism sacrifices some interpretability but provides significantly improved stability and scalability. In this **minimal design**, the structural stream is injected additively into hidden activations rather than directly modulating attention logits.

While the baseline perplexity grows explosively from **32.8 → 231.4** as sequence length increases from **256 → 12k tokens**, the bias-injected S-Former rises only from **29.6 → 50.86**, sustaining over

~**79% relative improvement** consistently across all lengths. Crucially, the degradation rate of S-Former remains nearly flat in Figure 3, in stark contrast to the baseline's seven-fold blowup.

Together, these results show that a recurrent structural stream—whether fused into attention or injected as bias—significantly mitigates the dilution effect. In the **40k-token extreme stress test**, baseline perplexity explodes to **321.4**, whereas Fusion and Bias Injection remain far more stable at **152.9 and 68.5**, respectively, corresponding to 47.7% and 76.4% degradation reduction. This demonstrates that structural anchoring continues to provide substantial gains even under extreme extrapolation.

Table 2: **Perplexity (PPL) extrapolation on WikiText-103 up to 40k tokens.** Both baseline and S-Former are trained only on 256† tokens.

| Length | Baseline PPL | Fusion PPL | Bias PPL |
|---|---|---|---|
| 256† | 32.84 | 29.87 | 29.63 |
| 512 | 35.42 | 29.45 | 28.68 |
| 2194 | 105.22 | 48.11 | 32.91 |
| 3876 | 152.75 | 60.06 | 36.72 |
| 5558 | 177.89 | 68.14 | 39.66 |
| 7241 | 195.11 | 74.97 | 41.94 |
| 8923 | 208.14 | 80.64 | 44.00 |
| 10605 | 219.98 | 85.79 | 45.64 |
| **12288** | **231.43** | **91.22** | **50.86** |
| 16384 | 248.84 | 103.11 | 54.60 |
| 20480 | 267.53 | 114.48 | 58.07 |
| 24576 | 279.83 | 123.49 | 61.31 |
| 28672 | 297.10 | 131.83 | 63.97 |
| 32768 | 306.90 | 139.19 | 66.54 |
| 36864 | 317.90 | 146.40 | 68.46 |
| **40960** | **321.40** | **152.96** | **68.50** |

Unlike attention fusion, bias injection does not expose a direct $\alpha$–structure correlation, since the structural stream is added additively into hidden activations rather than multiplicatively modulating queries and keys. As a result, interpretability is reduced. Nevertheless, complementary diagnostics demonstrate that bias injection continues to provide structural guidance. Layer-wise analysis (Table 4) shows a clear division of labor: the first layer functions as a structural parser with a high bias ratio (67.5%), subsequent layers refine structure with moderate bias ratios (43.0%, 38.1%), and deeper layers transition to semantic processing with a low bias ratio (24.6%). This quantitative pattern indicates that bias injection preserves structural anchoring implicitly, shifting structural responsibility to early layers while allowing deeper layers to focus on semantics. Orthogonality and temporal consistency further confirm that the injected structural stream remains disentangled and stable across layers. Thus, while attention fusion provides explicit $\alpha$–depth correlations as evidence of structure–content decoupling, **bias injection achieves the same structural guidance through implicit anchoring mechanisms**, making it more scalable and stable in practice.

## 4.3 LONG-CONTEXT REASONING: PG-19 AND LAMBADA

**Experimental Design and Dataset Selection.**

We focus on generative language modeling tasks rather than classification benchmarks for two reasons. First, our models are trained from scratch on limited data, making them unsuitable for knowledge-intensive tasks such as ARC-E/C (Clark et al., 2018) that assume extensive pre-training. Second, generative tasks directly test sequential modeling, which structural anchoring is designed to improve. For long-context evaluation, we use PG-19 (Rae et al., 2019) for book-length extrapolation and LAMBADA (Paperno et al., 2016) for dependency reasoning without external knowledge, ensuring that performance differences reflect architectural design rather than pretraining.

Table 3: **Performance on LAMBADA and PG-19.** S-Former consistently improves perplexity (PPL) and accuracy (ACC) over the baseline.

| | Baseline | | S-Former | |
|---|---|---|---|---|
| | PPL | ACC | PPL | ACC |
| **LAMBADA (last-token)** | | | | |
| Validation | 612.1 | 8.38 | 337.7 | 13.6 |
| Test | 620.1 | 7.71 | 338.7 | 12.8 |
| **PG-19 (token-level)** | | | | |
| Validation | 61.1 | 28.7 | 51.8 | 31.0 |
| Test | 55.8 | 30.1 | 47.1 | 32.4 |

**Summary.** Across PG-19 and LAMBADA, structural anchoring consistently improves long-context modeling, yielding both more stable extrapolation and higher dependency reasoning accuracy.

### 4.4 ABLATION STUDIES AND MECHANISM ANALYSIS

Beyond validating individual components, our ablations provide insight into why structural anchoring works. We analyze three key questions: (1) which components are necessary, (2) how the structural stream operates internally, and (3) what evidence supports structure vs. memory operation.

#### 4.4.1 INTEGRATION STRATEGY COMPARISON

Section 4.2 already showed that bias injection outperforms both vanilla Transformers and attention fusion. Here we briefly contrast their behaviors:

**Training Stability.** Attention fusion often drives $\alpha$ to 0/1, causing gradient issues and unstable convergence. Bias injection, by additive integration, avoids saturation and trains smoothly.

**Long-context Degradation.** Fusion improves over vanilla but still degrades beyond 10k tokens. Bias injection keeps degradation nearly flat, extrapolating stably up to 40k tokens.

**Interpretability vs. Scalability.** Fusion offers interpretable $\alpha$–depth correlations but suffers from instability in natural text. Bias injection trades some interpretability for robust and scalable performance.

Overall, attention fusion serves as a useful probe, but bias injection is the more practical design, balancing stability, scalability, and efficiency.

#### 4.4.2 COMPONENT CONTRIBUTION ANALYSIS

We ablate key components of bias injection to assess their roles:

**Structural Stream Removal.** Collapsing to a vanilla Transformer causes sharp degradation beyond 4k tokens, confirming the stream is essential for anchoring long contexts.

**No Gating ($\alpha = 1$) / No Regularization.** Without adaptive gating, all layers act as uniform injectors, reducing accuracy and eliminating layer-wise specialization. Similarly, removing the entropy-style penalty causes $\alpha$ to saturate at 0/1, destabilizing training and degrading extrapolation.

Overall, stable long-context behavior emerges from the synergy of (i) the structural stream, (ii) adaptive gating, and (iii) regularization.

#### 4.4.3 BIAS INJECTION MECHANISM ANALYSIS

To understand why bias injection succeeds, we probe its internal dynamics across layers (All metrics used in this analysis are formally defined in Appendix §B.1).

Table 4: **Layer-wise analysis of bias injection mechanism.** Each layer shows distinct functional roles, with early layers acting as structural parsers/refiners and deeper layers focusing on semantic processing. Bias ratio, orthogonality, and temporal consistency quantify the contribution of the structural stream across depth.

| Layer | Function | Bias ratio (%) | Orthogonality | Temporal consistency |
|-------|----------|----------------|---------------|---------------------|
| 0 | Structural Parser | 67.5 | 0.946 | 0.338 |
| 1 | Structural Refiner | 43.0 | 0.923 | 0.540 |
| 2 | Structural Refiner | 38.1 | 0.937 | 0.573 |
| 3 | Semantic Processor | 24.6 | 0.945 | 0.603 |

**Layer-wise Specialization.** Table 4 reveals a clear functional gradient across layers: bias injection enables the network to self-organize into specialized processing stages, with early layers focusing more on structural parsing (higher bias ratios) and deeper layers transitioning to semantic processing. This specialization emerges naturally from training, demonstrating that additive bias allows optimal division of labor without explicit programming.

**Orthogonal Representation.** Consistent with our definition of structure as a near-orthogonal subspace, the injected bias remains highly orthogonal to content (0.92–0.95) while utilizing nearly the full representational capacity ($\sim$200/256 effective rank). Spectral probes further show that bias injection expands the usable representation space: effective dimensionality improves by **1.43$\times$**, participation ratio by **1.77$\times$**, and spectral entropy by **1.09$\times$**. These results indicate that the structural stream contributes complementary, high-dimensional guidance rather than competing for semantic resources.

**Memory vs. Structural System.** A key concern is that bias injection might act as a disguised memory. Our evidence indicates otherwise:

1. **High dimensionality ($\sim$200 rank; +1.43$\times$ effective dimension) vs. compressed storage**: memory systems compress patterns into low-dimensional codes; our stream preserves near-full rank and even expands it, consistent with transformation rather than storage.
2. **Orthogonal operation (0.92–0.95) vs. overlap during retrieval**: memory retrieval typically overlaps with content vectors; our persistent orthogonality and steady rank utilization indicate complementary operation.
3. **Continuous evolution (0.34 $\to$ 0.60 temporal consistency) vs. discrete recall events**: memory retrieval produces episodic spikes; our smooth temporal progression reflects continuous coordinate refinement.

Together, these findings validate our view of structure as a dynamic, orthogonal subspace supporting the interpretation that bias injection contributes structural anchoring rather than acting as a compressed memory cache.

## 5 CONCLUSIONS, LIMITATIONS, AND FUTURE DIRECTIONS

We introduced the **Structural-Former (S-Former)**, which decouples content and structure through a recurrent structural stream. This stream provides persistent anchors that mitigate the dilution effect and enable stable long-context extrapolation. We explored two integration mechanisms: **attention fusion**, which offers interpretability but limited stability, and **bias injection**, a lightweight design that scales to 40k tokens. Our analyses indicate that the structural stream realizes *structural anchoring*, providing complementary high-dimensional guidance that improves long-context stability in Transformers. Overall, our results demonstrate that structural anchoring offers a practical path toward more stable long-context Transformers without altering their asymptotic complexity. While our study focuses on moderate model sizes, the principles of structural anchoring are directly compatible with larger-scale LLMs, suggesting a promising direction for future long-context architectures. These findings underline that even modest architectural changes, when guided by structural priors, can yield stable long-context extrapolation and remain compatible with scaling to larger models.

**Limitations.** S-Former introduces an additional recurrent stream alongside attention and feed-forward layers. The update is lightweight—linear in sequence length and parameter count ($< 10\%$ under our settings)—and does not alter the dominant $O(L^2 d)$ attention complexity. In practice the design trains stably without increasing memory or wall-clock bottlenecks. The main limitation is interpretability: bias injection is robust but its contribution is less directly visible than attention fusion. Another open point is theory: while our probes support structural anchoring empirically, its formal underpinnings remain to be fully developed.

**Future directions.** An important extension is to design **parallel-friendly variants** of the structural stream, for example by adapting scan-based or state-space formulations, so that structural anchoring fully aligns with modern large-scale accelerators. Another promising direction is **hybrid designs** that combine local recurrent anchoring with global parallel attention, balancing stability and efficiency across scales. Finally, evaluating structural anchoring in the context of large language models (LLMs) is a natural next step to assess scalability and practical impact.

## REPRODUCIBILITY STATEMENT

We have provided detailed descriptions of model architectures, training setups, datasets, and evaluation protocols in the main text and appendix. All baselines are implemented with parameter-matched settings for fair comparison. To ensure reproducibility, we will release our full codebase, training scripts, and data generation pipelines upon acceptance of this paper, along with instructions to reproduce all reported experiments.

## ETHICS STATEMENT

This work focuses on methodological contributions to long-context modeling and does not involve any private or personally identifiable data. All datasets used in our experiments (WikiText-103, PG-19, LAMBADA, Dyck, JSON, and Markov probes) are publicly available and widely used in the community. We do not foresee immediate ethical concerns from our experiments. However, as with other large language models, potential misuse for generating misleading or harmful content is possible. We encourage responsible use and release our methods and models solely for research and educational purposes.

## LLM USAGE

Large language models (LLMs) were used solely as assistive tools for polishing the writing of this paper (e.g., improving clarity, grammar, and readability). No parts of the research ideation, experimental design, implementation, or analysis were conducted by LLMs. All scientific contributions and experiments were carried out entirely by the authors.

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

# APPENDIX

## A EXPERIMENTAL RESULTS

### A.1 MARKOV PROBES

We design a toy Markov transition task to test whether attention can recover structured dynamics. A sparse block-graph is converted into a row-stochastic matrix $P^\star$. Each training sample is a sequence $[\texttt{0..V-1, } i]$ where the last token $i$ is the query, and the target distribution is $P^\star_{i,:}$. We compare a parameter-matched vanilla Transformer and an S-Former layer, training with KL divergence between the query's attention row and $P^\star_{i,:}$. Evaluation reports the mean $\ell_1$ distance between learned $\hat{P}$ and $P^\star$.

---

**Algorithm 1** Markov Attention Matching

---

1: Generate sparse graph $\to P^\star$
2: **for** epoch **do**
3:     **for** query $i$ **do**
4:         Input $[\texttt{0..V-1, } i] \to$ model
5:         Extract attention row at last position
6:         Minimize KL$(P^\star_{i,:} \| \hat{p}_{i,:})$
7: Eval: average rows $\to \hat{P}$, compute L1$(\hat{P}, P^\star)$

---

### A.2 DYCK PROBES

We evaluate S-Former on Dyck language completion tasks as a canonical probe for hierarchical generalization. **Data generation.** Sequences are sampled with three bracket types $()\,[]\,\{\}$ using

---

**Algorithm 2** Dyck Completion Training & Evaluation

---

1: **Input:** Generator $G$, model $M$, config $C$
2: Generate train/val/test via $G$
3: **for** epoch = 1..50 **do**
4:     **for** batch in train **do**
5:         $X \leftarrow$ sequence[:-1], $Y \leftarrow$ sequence[1:]
6:         $mask \leftarrow 1$ on target positions
7:         $logits \leftarrow M(X)$
8:         $L \leftarrow$ CE$(logits, Y$, ignore=$\texttt{<pad>})$
9:         $loss \leftarrow \sum(L \cdot mask) / \sum(mask)$
10:         Update $M$ with AdamW, gradient clip, scheduler
11: **Evaluation (greedy):** given $\texttt{<sos>}$+input, generate until $\texttt{<eos>}$ or max length; filter to $()\,[]$.
12: Compute Exact-Match, Structural Acc, Valid-but-not-Exact, Syntax Error.
13: **Extrapolation:** repeat for lengths $\{64, 80, 100\}$ and depths $\{3, ..., 8\}$.

---

a stack-based generator. Each completion sample is split into *(input, target)*, where the model sees $\texttt{<sos>}$+input and must generate the target. Depth-controlled sets ($d = 3$–$8$) are also constructed. Invalid samples are produced by deleting, replacing, or inserting random brackets.

**Training.** Models are trained with AdamW (lr $= 1e{-}4$), batch size 32, 50 epochs, and 3 warmup epochs followed by cosine decay. The loss is masked to only target tokens.

**Evaluation metrics.**

- Exact-Match Accuracy
- Structural Accuracy (valid Dyck sequence check)
- Valid-but-not-Exact rate
- Syntax Error rate $= 1-$ Structural Accuracy

**Extrapolation.** We evaluate on extended lengths ($L \in \{64, 80, 100\}$) and depths ($d = 3$–$8$). For S-Former, we additionally test sensitivity by corrupting the structural stream (permutation or noise).

## A.3 JSON PROBES

We evaluate S-Former on JSON serialization and completion to test hierarchical and schema-aware generalization.

**Data generation.** We synthesize JSON objects/arrays with controlled nesting depth and width. Keys are sampled from a fixed vocabulary (e.g., {"name","id","value","items","meta","ts"}), values are strings, integers, booleans, nulls, or recursively nested structures. Each example is converted to a compact, whitespace-free string; we then split it into *(input, target)* such that the model sees `<sos>`+input and must generate the remaining *target*. Negative (invalid) samples are created by bracket/quote corruption, missing commas/colons, or reordering that breaks JSON syntax. A validator based on `json.loads` defines structural validity.

**Controls.** We construct depth-controlled sets ($d = 2$–$6$), width-controlled sets (avg keys per object $w = 2$–$8$), and length buckets (token length $L \in \{128, 256, 512\}$). Key-set splits ensure that some keys only appear at test time to probe schema extrapolation.

**Training.** AdamW (lr $= 1e{-}4$), batch size 32, 50 epochs, 3 warmup epochs then cosine decay. Loss is masked to target tokens only.

**Evaluation metrics.**

- **Structural Validity**: passes `json.loads` (*valid JSON*).
- **Field F1**: compare parsed objects on a normalized key set (micro-F1 over present/absent key paths).
- **Syntax Error rate** $= 1-$ Structural Validity.

| Model | Struct (loose) | Depth (loose) |
|---|---|---|
| Standard Transformer | 0.845 | 0.095 |
| RoPE Transformer | 0.857 | 0.125 |
| S-Former (`pure`) | 0.903 | 0.035 |
| S-Former (`fused`) | 0.806 | 0.100 |
| S-Former (`dynamic`) | **0.953** | **0.055** |

---

**Algorithm 3** JSON Completion Training & Evaluation

---

1: **Input:** JSON generator $G$, model $M$, config $C$
2: Sample JSON trees with depth/width controls; stringify without spaces
3: Split each string into *(input, target)*; build sequences with `<sos>`
4: **for** epoch = 1..50 **do**
5:    **for** batch in train **do**
6:       $X \leftarrow$ sequence[:-1], $Y \leftarrow$ sequence[1:]
7:       $mask \leftarrow 1$ on target positions; 0 elsewhere
8:       $logits \leftarrow M(X)$
9:       $L \leftarrow \text{CE}(logits, Y, \text{ignore=}\texttt{<pad>})$
10:      $loss \leftarrow \sum(L \cdot mask)/\sum(mask)$
11:      Update $M$ (AdamW, grad clip, scheduler)
12: **Evaluation (greedy):** given `<sos>`+input, generate until `<eos>` or max length
13: **Structural Validity:** `try:` `json.loads(generated)` $\Rightarrow$ valid/invalid
14: **Field F1:** parse gold/pred JSON; flatten to key-path sets; compute micro-F1
15: Report Exact-Match, Structural Validity, Field F1, Syntax Error
16: **Extrapolation:** repeat across depths $\{2..6\}$, lengths $\{128, 256, 512\}$, and unseen key-sets
17: **(S-Former only)** optionally corrupt structural stream (permute/noise) at eval to measure sensitivity

---

## A.4 WIKITEXT-103 SETUP (ATTENTION FUSION)

**Architecture.** Baseline: a Pre-LN Transformer with RoPE positional encoding (base = 50k) applied to $Q, K$; $d_{\text{model}} = 256$, $n_{\text{layers}} = 4$, $n_{\text{heads}} = 8$, $d_{\text{ff}} = 1024$, dropout=0.1. S-Former (*attention*

*fusion*): replace selected layers (default: all four) with structural-fusion blocks; the fused representation feeds $Q, K$ while $V$ is taken from the content path. All other components remain identical.

**Training.** Models are trained only on 256-token windows with a sliding window (stride 64; 75% overlap). We use AdamW (1e−4), 2k warmup followed by cosine decay, batch size 64, label smoothing 0.03, and gradient clipping at 1.0.

**Evaluation.** We evaluate on the standard WikiText-103 test split and extrapolate to long contexts up to $L = 40{,}960$, reporting PPL/BPB and degradation relative to $L = 256$.

**Note.** This subsection reports the *attention fusion* integration. The *bias injection* variant uses the same data, model size, and schedule; only the integration mechanism differs (see the next subsection).

### A.5 WikiText-103 Setup (Bias Injection)

**Architecture.** The baseline is a 4-layer Pre-LN Transformer (256 hidden size, 8 heads, 1024 FFN, dropout 0.1) with RoPE on queries and keys. Bias-injection S-Former augments each layer with a lightweight structural stream implemented as a GRU pathway. The structural state is added as a gated bias into hidden activations before both attention and feedforward blocks. Gating values are regularized to remain within a stable range, with a simple warm-up schedule applied during training.

**Training.** Both baseline and S-Former are trained on 256-token windows with stride 64, using AdamW (lr $1 \times 10^{-4}$, 2k warmup, cosine decay), batch size 64, label smoothing 0.03, and gradient clipping at 1.0.

**Evaluation.** PPL/BPB are reported on the WikiText-103 test split for $L = 256$ up to 40k tokens, with smaller batch sizes at long contexts. We also report gate statistics (mean $\alpha$ and near-saturation ratios).

**Contrast.** Unlike *attention fusion*, which mixes content and structure into $Q, K$, bias injection preserves the content stream and injects $g_t$ as an additive memory bias at attention and FFN inputs.

## B  Implementation Details for LAMBADA and PG-19

**Common Training Framework.** Both LAMBADA and PG-19 experiments share the same implementation backbone. We use a 4-layer Pre-LN Transformer baseline ($d_{\text{model}} = 256$, $n_{\text{heads}} = 8$, $d_{\text{ff}} = 1024$, dropout $= 0.1$) with RoPE (base $= 10^5$) applied to queries and keys. For S-Former, we replace designated layers with structural blocks (GRU-based structural stream, bias injection, and $\alpha$-gating with warmup scheduling). Training uses AdamW with cosine decay and 2k warmup steps, batch size 16–32, stride $\approx L/4$, label smoothing 0.03–0.05, gradient clipping 1.0, and gate warmup 4k steps with $\tau$ annealed from $4 \rightarrow 2$. All runs are trained for 5–10 epochs on consumer GPUs (A100), and we report mean values across multiple seeds when available.

**Evaluation Metrics.** Perplexity (PPL) is computed from negative log-likelihood; Bits-per-byte (BPB) is also reported for extrapolation experiments. Accuracy (ACC) differs slightly:

- **LAMBADA**: last-token accuracy, i.e., whether the model predicts the final word correctly.
- **PG-19**: token-level accuracy, i.e., the fraction of all next-token predictions that are correct.

**Dataset Handling.** Both datasets are loaded locally when possible, falling back to HuggingFace repositories. Sliding-window segmentation is applied during training. For evaluation:

- **LAMBADA**: Only samples whose target is a single token are retained (standard preprocessing). Evaluation is performed at sequence lengths 512–4096.
- **PG-19**: Trained and evaluated at length 4096 tokens, without overlap (fixed blocks). This reflects the book-level nature of PG-19.

**Reproducibility.** We release both scripts as `lambada_bias.py` and `pg19_bias.py`. The codebases are identical up to ∼95%, differing only in dataset loader and accuracy metric. For transparency, we provide both scripts in the repository, but summarize them here in unified form.

## B.1 SUPPLEMENTARY PROBE ANALYSES (BIAS INJECTION)

We provide definitions of the probes used to analyze the bias stream $m_t = \alpha_t g_t$ in the bias injection variant.

- **Relative strength.** Ratio of bias to content norm:

$$r_t = \frac{\|m_t\|}{\|x_t\| + \varepsilon}.$$

- **Orthogonality.** Cosine-based independence measure:

$$\mathrm{Orth}(t) = 1 - \left|\langle \hat{m}_t, \hat{x}_t \rangle\right|.$$

- **Effective rank.** From SVD of bias vectors across tokens, the smallest $k$ explaining 95% variance.

- **Temporal consistency.** Cosine similarity between consecutive bias vectors:

$$\mathrm{TempCons} = \frac{1}{T-1} \sum_{t=1}^{T-1} \langle \hat{m}_t, \hat{m}_{t+1} \rangle.$$

- **Representation shift.** Change of hidden states with and without bias:

$$\Delta h_t = \frac{\|h_t^{\mathrm{biased}} - h_t^{\mathrm{orig}}\|}{\|h_t^{\mathrm{orig}}\| + \varepsilon}.$$

- **Spectral analysis.** Given eigenvalues $\{\lambda_i\}$ of the attention matrix:

$$\mathrm{EffDim} = \frac{\left(\sum_i \lambda_i\right)^2}{\sum_i \lambda_i^2}, \quad H = -\sum_i p_i \log(p_i + \varepsilon), \quad p_i = \frac{\lambda_i}{\sum_j \lambda_j}.$$

Here EffDim is effective dimensionality, and $H$ is spectral entropy.

Across layers, these probes show that the bias stream remains nearly orthogonal (0.92–0.95), high-rank ($\sim$200/256), and temporally smooth (0.34$\rightarrow$0.60), supporting its interpretation as a structural anchor rather than a compressed memory cache.

