# OpenReview forum: "S-Former: Structural Anchoring for Stable Long-Context Modeling"
_ICLR.cc/2026/Conference — ICLR 2026 Conference Withdrawn Submission_

### Official Review · Reviewer_zVko · 2025-10-25

**Soundness:** 3
**Presentation:** 3
**Contribution:** 3
**Rating:** 6
**Confidence:** 3

**Summary:**

This paper proposes the Structural-Former (S-Former), a Transformer variant that introduces a recurrent structural stream to stabilize long-context modeling. The key idea is to decouple content and structure by maintaining a parallel GRU-based structural pathway that evolves smoothly across tokens and provides “anchors” for attention. Two integration mechanisms are explored: attention fusion and bias injection. The model demonstrates strong improvements over vanilla Transformers on synthetic structure-sensitive benchmarks (Dyck, JSON, Markov transitions) and on long-context language modeling tasks (WikiText-103, PG-19, LAMBADA), showing significantly reduced perplexity degradation with increasing context length. Overall, this is an interesting contribution to long-context modeling. It proposes a simple yet effective inductive bias.

**Strengths:**

- I very much like this approach. In fact, I was considering exploring something similar in my own research — I’m happy to have been “scooped” in such an elegant way!
- The paper presents clear empirical advantages over standard Transformers across both synthetic and natural language tasks.
- The motivation is well articulated.
- The model is lightweight, conceptually simple, and appears to integrate seamlessly into standard Transformer stacks, suggesting strong potential for practical adoption.
- The authors provide multiple probing analyses that support their interpretation.

**Weaknesses:**

- Even after reading the paper, it remains unclear why exactly the gating mechanism helps with long-context extrapolation. Some mechanistic or theoretical intuition would strengthen this substantially.
- Some claims are taken for granted - particularly the “dilution effect” in attention. The text cites a few works, but they do not clearly demonstrate dilution as defined here. Empirical evidence (e.g., attention entropy vs. sequence length) would make this claim more solid.
- Section 4.4, which discusses the mechanism and ablations, is somewhat terse and could benefit from additional visualizations (e.g. rank/orthogonality trajectories etc.)

**Questions:**

- Can the authors provide direct empirical evidence that standard attention exhibits the “dilution effect” (e.g., flattening of attention scores or rising entropy with length)?
- How does the proposed mechanism relate to attention sinks, which have been hypothesized to stabilize or sharpen attention (e.g., Barbero et al., 2025)? Is the structural stream implicitly creating stable anchors akin to those sinks?
- How does this design compare to the recent Qwen gating approach (Qiu et. al. 2025)
- What is the effect of structural bias injection on the magnitude and variance of residual activations? Does it mitigate or exacerbate massive activation problems?
- How does the effective rank of the structural and content streams evolve with depth? Can this be connected to recent findings on rank collapse and anisotropy in deep Transformers (e.g., Skean et al., 2025; Queipo-de-Llano et al., 2025)?

Barbero, Federico, et al. "Why do LLMs attend to the first token?." arXiv preprint arXiv:2504.02732 (2025).

Qiu, Zihan, et al. "Gated Attention for Large Language Models: Non-linearity, Sparsity, and Attention-Sink-Free." arXiv preprint arXiv:2505.06708 (2025).

Skean, Oscar, et al. "Layer by layer: Uncovering hidden representations in language models." arXiv preprint arXiv:2502.02013 (2025).

Queipo-de-Llano, Enrique, et al. "Attention Sinks and Compression Valleys in LLMs are Two Sides of the Same Coin." arXiv preprint arXiv:2510.06477 (2025).

---

### Official Review · Reviewer_ajSA · 2025-10-28

**Soundness:** 2
**Presentation:** 2
**Contribution:** 2
**Rating:** 2
**Confidence:** 3

**Summary:**

This article proposes a new approach to address a fundamental problem in Transformer based language models: the dilution of attention in long contexts. Introducing a recurrent structural flow as a lightweight orthogonal induction bias.

**Strengths:**

1. Novel. The idea of decoupling "content" and "structure" into parallel streams is a significant conceptual contribution. It directly addresses the core limitation of metric-only positional encodings.

2. Comprehensive Validation. From controlled synthetic probes (Markov, Dyck, JSON) to interpretable mechanisms (attention fusion) and finally to bias injection.

**Weaknesses:**

1. Lack of an important ablation experiment, which is GLU-only. In my understanding, using only glu without relying on attention architecture has the potential to achieve high accuracy in the Structural accuracy on Dyck benchmark shown in Table 1.

2. Lack of discussion on related work. From eq2-5, it can be seen that S-former is a model that combines no linear RNN and attention. Therefore, it will be necessary to discuss the related work about rnn-attention hybrid models.

3. The method is difficult to efficiently parallelize. At present, non-linear RNNs such as GLU cannot be efficiently implemented using GPUs. So I strongly doubt what line 471 said about "In practice the design trains stably without increasing memory or wall-clock bottlenecks. ". Therefore, providing speed comparisons on GPUs is necessary.

**Questions:**

See Weakness.

---

### Official Review · Reviewer_UfjR · 2025-11-02

**Soundness:** 2
**Presentation:** 2
**Contribution:** 3
**Rating:** 4
**Confidence:** 4

**Summary:**

This paper introduces S-Former, a modified Transformer that strengthens the structural information when modeling long sequences. It tackles the "dilution effect" where attention weakens over distance. The key idea is a recurrent "structural stream" that tracks the sequence's underlying structure (e.g., grammar, hierarchy) separately from its content. This structural signal is then injected into the Transformer as a simple bias.

**Strengths:**

- The paper addresses the critical challenge of modeling structural information in long sequences, a well-known limitation of standard Transformers. The core hypothesis of separating structural and semantic representations is intuitive.
- Using a recurrent stream in the attention mechanism is an active area of research. The proposed method, utilizing a classic GRU as a recurrent component, is a new contribution to this field.
- The analysis correlating the $\alpha$ gate in "attention fusion" with bracket depth in the Dyck language task (Fig. 2) is a clear and inspiring demonstration of the model learning an explicit structural property.

**Weaknesses:**

1. The experiments do not convincingly prove strong long-context capabilities. The baseline model's perplexity "explodes" (line 329) to around 300 at 40k tokens. As shown by prior work [1, 2], a well-trained LLM's perplexity typically rockets to 1000+ immediately upon extrapolating beyond its training context. The baseline's comparatively gentle degradation suggests it **may be undertrained** or that the **small model size and short training length** (256 tokens) create an experimental setup where improvements are easier to show. This calls into question whether the S-Former's gains would transfer to more realistic, large-scale models.
2. The evaluation relies heavily on perplexity (PPL), which has been shown to be an **unreliable indicator of long-context** understanding [3]. To strengthen its claims, the paper should include results on more standard, task-based long-context benchmarks such as Needle-in-a-Haystack (NIAH), LongBench, or RULER.
3. Without a **parallel training algorithm** (as seen in recent work like GLA [4] or Mamba), it is impractical to address long contexts and confines the experiments to a short training length of 256. The claim of minimal overhead (line 185, 471) is not well-supported without actual training wall time data.
4. The motivation for the confidence penalty (Eq. 8) is unclear. The authors state that bias injection already resolves the numerical stability issues of attention fusion (line 183). It seems redundant to then add the regularization term.
5. Section 4.4 makes several claims about the importance of different components (e.g., gating, regularization) but does not provide the corresponding **quantitative ablation results** to prove them. These arguments remain unsubstantiated without supporting data.
6. Key concepts like GRU (Eq. 2) should be defined, at least in the appendix, or cited. Other core concepts like the "dilution effect" and "attention span" are **used but never formally defined**. Even the "S-Former" architecture itself could be more precisely defined.
7. The **writing** needs to be improved. For instance, the summary of Section 4 (lines 194-198) feels out of place. The analysis in Section 4.4 might be better integrated into Section 3 (Methods) to explain the design choices, while the ablation section itself should focus on presenting quantitative results.
8. The **notations** are not clear. See questions for details.


[1] [Extending Context Window of Large Language Models via Positional Interpolation](https://arxiv.org/abs/2306.15595)
[2] [YaRN: Efficient Context Window Extension of Large Language Models](https://openreview.net/forum?id=wHBfxhZu1u)
[3] [Can Perplexity Reflect Large Language Model's Ability in Long Text Understanding?](https://openreview.net/forum?id=Cjp6YKVeAa&noteId=Cjp6YKVeAa)
[4] [Gated Linear Attention Transformers with Hardware-Efficient Training](https://arxiv.org/abs/2312.06635)

**Questions:**

1. Why was "attention fusion" used for the Dyck language probe (Sec 4.1.2) instead of the final proposed "bias injection" method? Could you clarify the trade-offs and explain why one was chosen over the other in that specific context? The method name is also confusing.

The notations are inconsistent and need clarification:

1. Clarify the shapes of all the parameters, states, including $W, g_t, \alpha_t$, etc.
2. Is $\alpha$ shared across layers or computed independently for each layer?
3. In Eq. 3, do the brackets ($[g_t, LN(x_t)]$) denote concatenation? Please state this explicitly.
4. In Eq. 4 and 6, you use $\alpha_t$. Should these be distinguished as fusion and bias superscripts to avoid confusion?

---

### Official Review · Reviewer_tS8t · 2025-11-03

**Soundness:** 2
**Presentation:** 2
**Contribution:** 2
**Rating:** 2
**Confidence:** 4

**Summary:**

This paper proposes the StructuralFormer (S-Former), a Transformer-based model that introduces a parallel recurrent structural stream to decouple content (semantic meaning) and structure (sequential/hierarchical arrangement) of sequences, aiming to mitigate the "dilution effect" in long-context modeling.

**Strengths:**

1. The proposed bias injection mechanism offers a practical trade-off between performance and efficiency.

2.The paper adopts a multi-stage validation pipeline, starting with controlled synthetic tasks to isolate structural learning, followed by interpretability analyses and large-scale natural language evaluations.

**Weaknesses:**

1. Equation 2 employs a GRU to compute the structural state, but the paper fails to provide a compelling rationale for why a GRU is suitable for extracting structural information. While the authors emphasize structure-content decoupling as a core motivation, the specific mechanism that enables this decoupling remains unclear. For instance, the paper mentions "near-orthogonality" between structural and content representations (Section 3.2) but does not explain how the GRU’s gated recurrence contributes to this orthogonality.

2.Using recurrent units to capture sequential dynamics in conjunction with Transformers is not new—prior works have already integrated recurrent mechanisms with Transformers for long-context modeling. Similarly, linear fusion of recurrent states with Transformer hidden states (as in the attention fusion mechanism, Equations 3–4) has been explored in earlier hybrid architectures.

3. The paper exhibits inconsistent notation that impairs readability. In Equation 3, \alpha is defined as the structure-content trade-off gate, but in Equation 4, this gate is incorrectly denoted as \alpha when modulating the query and key.

4. For Equation 6, the paper does not specify the number of additional parameters introduced by the bias injection mechanism. Even Section 4.4.3, which analyzes the bias injection mechanism, it omits critical details such as the dimensionality of the weight matrix.

**Questions:**

NA

---

### Note · Authors · 2026-01-04

**Comment:**

Thanks for all the comments! we are working on improving our hypothesis and experiments! See you all around!

**Withdrawal Confirmation:**

I have read and agree with the venue's withdrawal policy on behalf of myself and my co-authors.